# Early Routine Biomarkers of SARS-CoV-2 Morbidity and Mortality: Outcomes from an Emergency Section

**DOI:** 10.3390/diagnostics12010176

**Published:** 2022-01-12

**Authors:** Flavio Maria Ceci, Marco Fiore, Francesca Gavaruzzi, Antonio Angeloni, Marco Lucarelli, Carolina Scagnolari, Enea Bonci, Francesca Gabanella, Maria Grazia Di Certo, Christian Barbato, Carla Petrella, Antonio Greco, Marco De Vincentiis, Massimo Ralli, Claudio Passananti, Roberto Poscia, Antonio Minni, Mauro Ceccanti, Luigi Tarani, Giampiero Ferraguti

**Affiliations:** 1Department of Experimental Medicine, Sapienza University of Rome, 00185 Roma, Italy; flaviomaria.ceci@uniroma1.it (F.M.C.); antonio.angeloni@uniroma1.it (A.A.); marco.lucarelli@uniroma1.it (M.L.); enea.bonci@uniroma1.it (E.B.); giampiero.ferraguti@uniroma1.it (G.F.); 2Institute of Biochemistry and Cell Biology (IBBC-CNR), Department of Sensory Organs, Sapienza University of Rome, 00185 Roma, Italy; francesca.gabanella@cnr.it (F.G.); mariagrazia.dicerto@cnr.it (M.G.D.C.); christian.barbato@cnr.it (C.B.); carla.petrella@cnr.it (C.P.); 3Department of Public Health and Infectious Diseases, Sapienza University of Rome, 00185 Roma, Italy; francesca.gavaruzzi@uniroma1.it; 4Laboratory of Virology, Department of Molecular Medicine, Istituto Pasteur Italia-Fondazione Cenci Bolognetti, Sapienza University of Rome, 00185 Roma, Italy; carolina.scagnolari@uniroma1.it; 5Department of Sensory Organs, Sapienza University of Rome, 00185 Roma, Italy; antonio.greco@uniroma1.it (A.G.); marco.devincentiis@uniroma1.it (M.D.V.); massimo.ralli@uniroma1.it (M.R.); antonio.minni@uniroma.it (A.M.); 6Institute of Molecular Biology and Pathology (IBPM-CNR), 00185 Rome, Italy; claudio.passananti@cnr.it; 7Unita di Ricerca Clinica e Clinical Competence-Direzione Generale, AOU Policlinico Umberto I, 00161 Roma, Italy; roberto.poscia@uniroma1.it; 8Società Italiana per il Trattamento dell’Alcolismo e le sue Complicanze (SITAC), 00184 Roma, Italy; mauro.ceccanti@uniroma1.it; 9Department of Pediatrics, Sapienza University of Rome, 00185 Roma, Italy; luigi.tarani@uniroma1.it

**Keywords:** COVID-19, SARS-CoV-2, emergency section, intensive care unit, mortality, morbidity, biomarker, early predictor

## Abstract

Background. COVID-19 is a severe acute respiratory disease caused by SARS-CoV-2, a virus belonging to the Coronaviridae family. This disease has spread rapidly around the world and soon became an international public health emergency leading to an unpredicted pressure on the hospital emergency units. Early routine blood biomarkers could be key predicting factors of COVID-19 morbidity and mortality as suggested for C-reactive protein (CRP), IL-6, prothrombin and D-dimer. This study aims to identify other early routine blood biomarkers for COVID-19 severity prediction disclosed directly into the emergency section. Methods. Our research was conducted on 156 COVID-19 patients hospitalized at the Sapienza University Hospital “Policlinico Umberto I” of Rome, Italy, between March 2020 and April 2020 during the paroxysm’s initial phase of the pandemic. In this retrospective study, patients were divided into three groups according to their outcome: (1) emergency group (patients who entered the emergency room and were discharged shortly after because they did not show severe symptoms); (2) intensive care unit (ICU) group (patients who attended the ICU after admission to the emergency unit); (3) the deceased group (patients with a fatal outcome who attended the emergency and, afterward, the ICU units). Routine laboratory tests from medical records were collected when patients were admitted to the emergency unit. We focused on Aspartate transaminase (AST), Alanine transaminase (ALT), Lactate dehydrogenase (LDH), Creatine kinase (CK), Myoglobin (MGB), Ferritin, CRP, and D-dimer. Results. As expected, ANOVA data show an age morbidity increase in both ICU and deceased groups compared with the emergency group. A main effect of morbidity was revealed by ANOVA for all the analyzed parameters with an elevation between the emergency group and the deceased group. Furthermore, a significant increase in LDH, Ferritin, CRP, and D-dimer was also observed between the ICU group and the emergency group and between the deceased group and ICU group. Receiver operating characteristic (ROC) analyses confirmed and extended these findings. Conclusions. This study suggests that the contemporaneous presence of high levels of LDH, Ferritin, and as expected, CRP, and D-dimer could be considered as potential predictors of COVID-19 severity and death.

## 1. Introduction

Severe acute respiratory syndrome coronavirus 2 (SARS-CoV-2) belongs to the Coronaviridae family and was initially identified in December 2019 in Wuhan, China [1]. Later, in February 2020, the World Health Organization (WHO) defined this infection as “coronavirus diseases 2019 (COVID-19)”, and on March 2020, it declared the COVID-19 pandemic [2]. As of 15 December 2021, more than 270 million cases have been reported across 192 countries, resulting in more than 5,312,314 deaths [3,4]. In Italy, the first two cases detected occurred in January 2020 in Rome when two Chinese tourists tested positive via the COVID-19 swab [5]. As of 15 December 2021, 5,238,221 positive cases were recorded in Italy, with 134,929 deaths and more than 178,000 active cases [3]. In this health crisis, clinical laboratory specialists search for reliable biomarkers associated with COVID-19 disease for the appropriate clinical management.

The SARS-CoV-2 virus is more likely to cause serious illness and a higher mortality rate in older people, though cases of COVID-19-related deaths have been even reported in young and middle-aged adults [6,7,8]. COVID-19 disease has been described through three phases [9]. During early infection, SARS-CoV-2 infects respiratory cells by interacting with the angiotensin-converting enzyme 2 (ACE2) receptor [10]. During the second phase, inflammation is localized in the lungs and leads to primarily respiratory symptoms, such as incessant cough and low blood oxygen saturation level [11]. The third phase is characterized by the cytokine storm, a condition that leads to acute respiratory distress syndrome, cardiovascular damage, and multiple organ failure [12,13]. Inflammatory markers such as interleukin-6 (IL-6), C-reactive protein (CRP), Procalcitonin (PCT), Lactate dehydrogenase (LDH), Ferritin, and D-dimer are highly increased during the cytokine storm, creating parenchymal lesions in vital organs [14].

When new infectious diseases break out, it is important to identify early biomarkers of morbidity and mortality in order to avoid progression to a severe form and death. This would help clinics select patients in the first phase of COVID-19 disease who are at higher risk of developing the worst prognosis. Timely and effective management of COVID-19 patients based on laboratory results is of great importance to optimize treatment management. Additionally, the ability to filter patients with severe disease from patients with mild disease would help reduce the workload and stress faced by emergency section professionals.

Most of the studies conducted during the first wave of the pandemic classified patients into mild, moderate, and severe groups according to the gravity of their signs and symptoms [15]. This was due to the great variability of clinical manifestations during the first phase of the infection [16]. In a previous study investigating COVID-19 survivors and deceased, the authors found significant differences between the two groups in terms of white blood cells, neutrophils, lymphocytes, monocytes, eosinophils, platelet counts, CRP, ferritin, and procalcitonin values from routine blood analysis [17]. The authors found a significant correlation between age, neutrophil/monocyte/platelet-to-lymphocyte ratio, and duration of hospital stay [17]. Accordingly, in the present study, we analyzed and compared routine biomarkers at the entrance to the emergency room of COVID-19-positive patients who subsequently developed a serious illness and spent time in the intensive care unit (ICU) or died to the values of patients attending only the emergency room who were discharged shortly thereafter for the absence of severe symptoms. Thus, the purpose of this retrospective study was to disclose, at the level of an emergency unit, those blood parameters that might serve as an early indication of a severe COVID-19 progression as observed for hospitalized patients [17,18,19,20,21]. We focused on Aspartate transaminase (AST), Alanine transaminase (ALT), Lactate dehydrogenase (LDH), Creatine kinase (CK), Myoglobin (MGB), Ferritin, CRP and D-dimer, all known to have a crucial role in COVID-19 onset and evolution and usually measured in the routine blood parameters of the Policlinico Umberto I hospital emergency section. We predicted that not only should CRP and D-dimer be considered as predictors of COVID-19 fatal progression, but other parameters should be contemporaneously taken into consideration for early intervention with appropriate treatments.

## 2. Materials and Methods

### 2.1. Participants’ Selection and Study Design

The study was conducted on 156 COVID-19 patients hospitalized at the Sapienza University Hospital “Policlinico Umberto I” of Rome, Italy, between March 2020 and April 2020. No vaccines were available at that time. We collected data from patients with SARS-CoV-2 infection who had been hospitalized. Patients were divided into three groups according to their outcome. The first group includes those patients who entered the emergency room and were discharged shortly after because they did not show severe symptoms (*emergency* group). The second group includes those patients admitted to the emergency room and then transferred to the COVID intensive care units who survived (*ICU* group). The third group includes those patients who were admitted to the emergency room and then transferred to the COVID intensive care units and died (*deceased* group). The diagnosis of SARS-CoV-2 infection was based on a positive result from real-time reverse-transcription polymerase chain reaction (RT-PCR) testing of nasopharyngeal-swab specimens. Patients who tested positive for the molecular test during recovery were transferred to the hospital’s COVID-19 wards. We included only positive SARS-CoV-2 individuals showing symptoms such as fever, dyspnea, cough, nausea, anosmia, or dysgeusia. However, some positive patients displayed arthralgia and asthenia.

The University Hospital ethical committee approved the study (Rif. 6536), and all the study procedures followed the Helsinki Declaration of 1975, as revised in 1983, for human rights and experimentations.

### 2.2. Data Collection

For each eligible patient, we extracted information on demographic characteristics (age and sex) and laboratory analytical results. The results of the laboratory tests were collected when patients were initially admitted to the emergency unit. Laboratory analytical results included liver biomarkers (AST and ALT)], LDH, cardiac biomarkers (CK and MGB), inflammatory biomarkers (Ferritin and CRP), and D-dimer. As further exclusion criteria, we only enrolled individuals with medical records including these laboratory analytical data (emergency group *n* = 31, 12 males and 19 females; ICU group *n* = 54, 26 males and 28 females; deceased group *n* = 71, 35 males and 36 females).

### 2.3. Laboratory Examination

The patients’ peripheral blood was collected in vacutainer tubes for blood testing. The BCS XP System automatic hemostatic analyzer (Siemens Healthcare, Erlangen, Germany) analyzed coagulation parameters such as D-dimer (reference range: 50–420 μg/L) using, respectively, immunological and Clauss modified methods. The interassay coefficient of variation (CV) was, respectively, 7.9% at a serum D-dimer of 200 μg/L. Tissue biomarkers of damage include AST (reference range: 9–45 U/L), ALT (reference range: 10–40 U/L), CK (reference range: 20–220 U/L), Ferritin (reference range: male 30–400 μg/L; female 15–150 μg/L), CRP (reference range: 100–6000 μg/L), LDH (reference range: 135–225 U/L) were measured using the standard colorimetric and enzymatic method on a Cobas C 501 analyzer (Roche Diagnostics, Germany). CV was, respectively, 2.3% at a serum AST of 30 U/L, 2.6% and at a serum ALT of 24 U/L, 3.2% at a serum CK of 18.7 U/L, 2.8% at a serum Ferritin of 26.1 μg/L, 1.3% at a serum CRP of 39.9 μg/L, and 2.7% at a serum LDH of 124 U/L. MGB (reference range: 28–72 μg/L) was measured on a Cobas E 601 analyzer, using sandwich immunological methods (Roche Diagnostics, Mannheim, Germany). The interassay CV was, respectively, 1.9% at a serum Myoglobin of 60.5 μg/L.

### 2.4. Statistical Analysis

According to methods previously described [22,23], data were analyzed to assess normality by Pearson’s chi-squared test and two-way analysis of variance (ANOVA). The emergency vs. ICU vs. deceased and males vs. females groups were used to analyze the laboratory parameters. Post hoc comparisons were carried out by using the Tukey’s HSD test. The Spearman Correlation test was used to investigate the correlation between the laboratory data and the age of the patients. A receiver operating characteristic (ROC) analysis was performed to measure the diagnostic/predictive accuracy of each variable [23].

## 3. Results

The age of the enrolled positive SARS-CoV-2 individuals is shown in Figure 1. The age range was 26–73 years (57.12 ± 2.03) for the emergency group, 19–88 years (66.85 ± 2.22) for the ICU group and 17–95 years (68.18 ± 1.58) for the deceased group. ANOVA data show an effect of morbidity (F(2150) = 7.53, *p* < 0.01) because of the higher values of both ICU and deceased groups compared with the emergency group (see the right upper panel of the figure, ps < 0.05 in post hocs). Data also showed mainly a gender effect (F(1150) = 4.43, *p* < 0.05) with a mild age elevation in positive women. No interaction morbidity vs. age was disclosed by ANOVA (F(2150) = 5.29, *p* = 0.074).

Figure 2 and Figure 3 show the laboratory data. In particular, AST, ALT, LDH and MGB are shown in Figure 2 whereas CK, Ferritin, D-dimer and CRP and shown in Figure 3. Quite interestingly, a main effect of morbidity was revealed by ANOVA for all the analyzed parameters (F(2150) = 3.69, 7.38, 21.26, 3.88, 3.30, 12.46, 18.98, 14.17, respectively, (AST, ALT, LDH, MGB, CK, Ferritin, D-dimer, CRP); ps < 0.05). Post hocs are shown in the right upper panels of each laboratory parameter. Indeed, the main differences emerged between the emergency group and the deceased group for AST, ALT, LDH, MGB, CRP, Ferritin, and D-dimer (ps < 0.05 in post hocs). A significant increase in LDH, CRP and D-dimer was also displayed between the ICU group and the emergency group (ps < 0.05 in post hocs). Finally, the values of LDH, Ferritin, and D-dimer of the deceased group were also higher compared with the values of the ICU group (ps < 0.05 in post hocs).

No main effects of gender emerged in the statistics for all the analyzed parameters (F(1150) = 0.07, 0.06, 0.67, 0.20, 0.19, 0.06, 0.05, 1.52, respectively, (AST, ALT, LDH, MGB, CK, Ferritin, D-dimer, CRP); ps not significant), and no interactions based on gender x morbidity were revealed (F(2150) = 0.43, 0.28, 0.07, 0.20, 0.71, 0.90, 0.32, 0.40, respectively, (AST, ALT, LDH, MGB, CK, Ferritin, D-dimer, CRP); ps not significant).

No effects of age were evidenced by the Spearman Correlation test in the analyzed parameters for all groups. In particular, Table 1 shows the Spearman Correlation for the deceased group.

Table 2 and Table 3 show the ROC data. The area under the curve (AUC) scores for LDH, MGB, CPR, Ferritin, and D-dimer evidence the highest scores (in bold in Table 2) in the deceased group. As for the ICU group, we found similar outputs but not for LDH. ALT crucial scores were also disclosed in both groups.

The positive predictive values (PPV) and the negative predictive values (NPV) based on the upper reference values for AST, ALT, LDH, MGB, CK, CRP, Ferritin, and D-dimer are shown in Table 3. In the deceased group, the highest PPV scores were evidenced for CRP and D-dimer and for Ferritin in women. The highest PPV scores of the ICU group were also the CRP, D-dimer, and Ferritin in women. Quite interestingly, the NPV highest values of the emergency group were found for AST, ALT, MGB, CK, CRP, and Ferritin (for women).

## 4. Discussion

In this study, we evaluated the levels of routine laboratory markers at the entrance to the emergency unit in a population of 156 COVID-19 patients who developed a poor prognosis reaching ICU or facing death. To do this, we examined a control population of COVID-19 patients who were initially admitted to the emergency unit. Subsequently, they were discharged because they did not show severe signs and symptoms. Our study shows that upon entrance to the emergency unit, patients who will develop a worse course have elevated levels of a variety of laboratory biomarkers widely used in the emergency room. ANOVA data revealed that LDH, Ferritin, CRP, and D-dimer levels were markedly elevated in deceased and ICU patients compared to the emergency group as previously observed for hospitalized patients [19,20,21]. In particular, the AUC scores suggest that elevations in a combination of blood parameters (LDH, MGB, CPR, Ferritin and D-dimer) at the emergency section level might lead to severe (ICU and/or death) health outcomes [18,23,24]. The PPV data also confirm and extend these findings, disclosing that a potentiation in Ferritin values should be carefully evaluated in women positive to COVID-19 [25,26,27]. A further indication of the present study regards the NPV scores corroborating that early normal range blood parameters (AST, ALT, MGB, CK, CRP, Ferritin for women, LDH) in COVID-19 patients could lead to mild or light consequences.

The strength and novelty of this investigation were to classify the COVID-19 patients according to their final prognosis instead of classifying patients using a score based on criteria correlated to severe COVID-19 disease. Indeed, the present retrospective study is focused at the level of the *emergency section only* to predict severe COVID-19 outcomes early by comparing three different groups of patients (those attending only the emergency section; those entering an emergency section and attending the ICU for COVID-19 after; those entering an emergency section and attending the ICU after but with a fatal outcome). In order to identify poor outcomes, similar studies were certainly performed but with different schedules and groups of patients [28,29]. Usually, the main criteria used are age, fever, respiratory rate, respiratory distress, oxygen saturation levels, arterial blood oxygen partial pressure, and the presence of bilateral and peripheral ground-glass opacities [30]. The limit of this approach is the great variability of the clinical manifestations of COVID-19 patients, which do not always correlate attentively with a poor prognosis [31]. On the contrary, in this study, we evaluated early laboratory biomarkers of patients who developed a poor prognosis because, after the first admission to the emergency unit, they ended up in the ICU or died.

Few studies have been conducted in the emergency unit regarding the study of early routine biomarkers for the prediction of morbidity and mortality of COVID-19 disease [28,29,32,33]. D-dimer is a product of fibrinolysis widely used as a marker of activation of the coagulation and fibrinolytic systems and for the diagnosis of thromboembolism [34]. High levels of D-dimer found in the severe form of COVID-19 disease indicate the clotting alterations typical of this disease. In particular, pulmonary micro-thrombosis and disseminated intravascular coagulation are common complications and causes of death [35]. Among inflammatory biomarkers, CRP, Ferritin, and LDH are some of the main biomarkers responsible for the hyperinflammatory response, which leads to acute respiratory distress syndrome and multiple organ failure [36]. CRP is an annular (ring-shaped) pentameric protein found in blood plasma whose circulating concentrations rise in response to inflammation. It is an acute-phase protein of hepatic origin that increases following interleukin-6 secretion by macrophages and T cells. [37]. In physiological conditions, blood Ferritin is present in small amounts, and it is an indirect indicator of iron storage in the organism [38]. In the case of acute Ferritin elevation, as in hyperferritinemia and COVID-19, it has been suggested that the combination of hyperferritinemic syndromes and long-lasting effects of inflammation can contribute to multiorgan disease, which characterizes the long-COVID-19 condition [39,40]. Both are acute-phase proteins used in clinical chemistry as inflammation markers. LDH is an intracellular enzyme localized in nearly all organ systems, which catalyzes the interconversion of pyruvate and lactate and the interconversion of NADH and NAD^+^ [41]. Severe forms of COVID-19 cause severe tissue damage and the subsequent release of large amounts of LDH into the circulation, indicating a severe form of interstitial pneumonia [42]. Our data displays how D-dimer, CRP, Ferritin and LDH could represent important routine biomarkers of COVID-19 disease, as shown in previous studies [17,24,43,44].

Indeed, the early identification of COVID-19 at-risk patients during the first phase of the disease using routine biomarkers might be crucial to prevent the development of acute respiratory failure and multi-organ damage. The outcomes of our study clearly show that the contemporaneous increase in the levels of some biomarkers for patients already in the emergency section was significantly associated with the severity of the disease. Based on the results of our study, we believe that the levels of D-dimer, CRP, ferritin, and LDH might help identify those COVID-19 patients at greater risk of developing a worse prognosis earlier, even in individuals showing no severe symptoms and signs in the emergency room. This study has limitations because it was conducted at the beginning of the pandemic. Indeed, gathering complete information to include in the medical records was difficult and complex. For this reason, many biomedical parameters are missing, especially for the emergency group individuals.

## 5. Conclusions

Biomarkers can be useful tools for clinicians in managing COVID-19 patients since biomarkers can reflect the need to begin an appropriate treatment early. We do believe that the combination of these data with epidemiological data may be useful in the diagnosis and management of COVID-19 diseases and future pandemics caused by SARS-CoV(n). The present study may also be considered a further step in the attempt to unravel early biomolecular indicators of COVID-19 progression, although differences may emerge between the several virus variants, i.e., the common Alpha and Delta variants and the recent Omicron. These findings may be of interest for studies in the fields of human disorders caused by viral or bacterial infections.

## Figures and Tables

**Figure 1 diagnostics-12-00176-f001:**
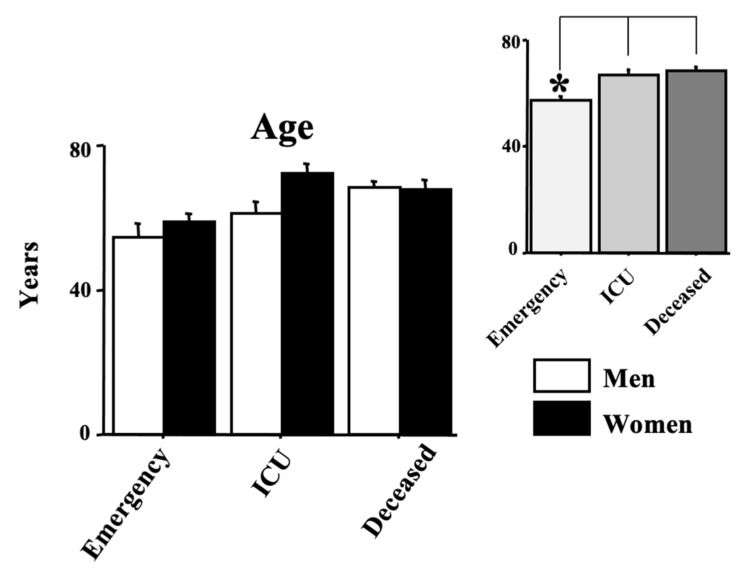
Age in years in COVID-19-positive individuals. Patients were divided into three groups according to their outcome: (1) emergency group (patients who entered the emergency room and were discharged shortly after because they did not show severe symptoms); (2) intensive care unit (ICU) group (patients who attended the ICU after admission to the emergency unit); (3) the deceased group (patients with a fatal outcome who attended the emergency and, afterward, the ICU units). The error bars indicate pooled standard error means (SEM) derived from the appropriate error mean square in the ANOVA. The asterisks indicate significant differences between groups in post hocs (* *p* < 0.05). The right upper panels show the data expressed only for the COVID-19 morbidity condition.

**Figure 2 diagnostics-12-00176-f002:**
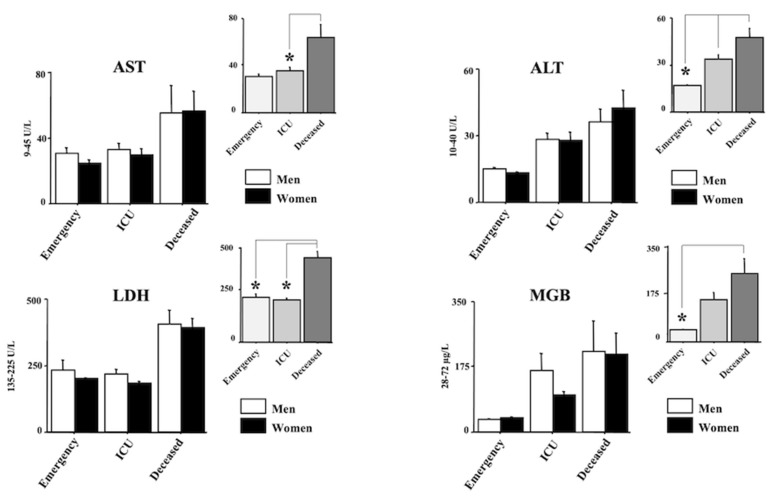
Aspartate transaminase (AST), Alanine transaminase (ALT), Lactate dehydrogenase (LDH), and Myoglobin (MGB) in positive individuals for COVID-19. Patients were divided into three groups according to their outcome: (1) emergency group (patients who entered the emergency room and were discharged shortly after because they did not show severe symptoms); (2) intensive care unit (ICU) group (patients who attended the ICU after admission to the emergency unit); (3) the deceased group (patients with a fatal outcome who attended the emergency and, afterward, the ICU units). The error bars indicate pooled standard error means (SEM) derived from the appropriate error mean square in the ANOVA. The asterisks indicate significant differences between groups in post hocs (* *p* < 0.05). The right upper panels show the data expressed only for the COVID-19 morbidity condition.

**Figure 3 diagnostics-12-00176-f003:**
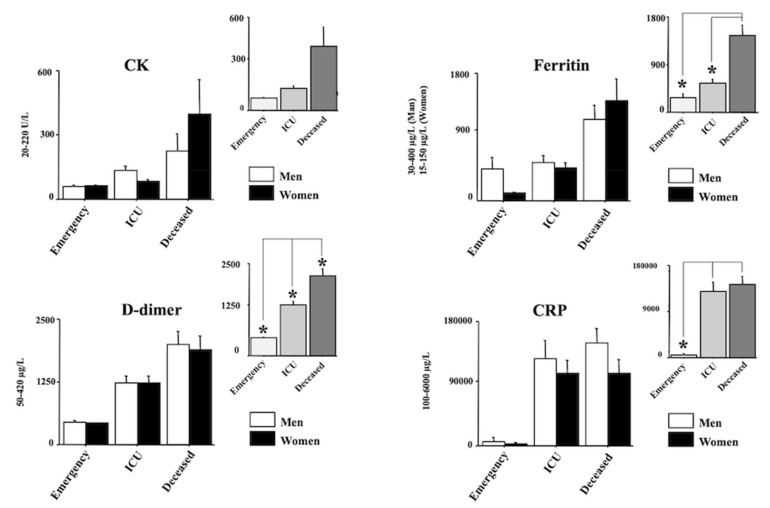
Creatine kinase (CK), Ferritin, D-dimer, and C-reactive protein (CRP), in positive individuals for COVID-19. Patients were divided into three groups according to their outcome: (1) emergency group (patients who entered the emergency room and were discharged shortly after because they did not show severe symptoms); (2) intensive care unit (ICU) group (patients who attended the ICU after admission to the emergency unit); (3) the deceased group (patients with a fatal outcome who attended the emergency and, afterward, the ICU units). The error bars indicate pooled standard error means (SEM) derived from the appropriate error mean square in the ANOVA. The asterisks indicate significant differences between groups in post hocs (* *p* < 0.05). The right upper panels show the data expressed only for the COVID-19 morbidity condition.

**Table 1 diagnostics-12-00176-t001:** Spearman Correlation values for the age parameter in the deceased group.

	Men			Women		
	SSD	Rho	*p*-Value	SSD	Rho	*p*-Value
**AST**	7992.5	−0.129	0.4863	6116.5	0.209	0.2080
**ALT**	7838	−0.108	0.5687	5484	0.291	0.0818
**LDH**	9200	−0.229	0.0925	6216	0.196	0.2367
**MGB**	8082	−0.141	0.4417	8711	−0.128	0.4737
**CK**	6934.5	0.021	0.8667	8583.5	−0.110	0.5357
**PCR**	7459	−0.053	0.7945	9495.5	−0.228	0.1889
**Ferritin**	7257	−0.024	0.9239	7088.5	0.084	0.6038
**D-dimer**	5955	0.159	0.3332	7397	0.043	0.7764

**Table 2 diagnostics-12-00176-t002:** AUC scores for AST, ALT, LDH, MGB, CK, CRP, Ferritin, and D-dimer (see methods). The highest scores (in bold) were disclosed for LDH, MGB, CPR, Ferritin, and D-dimer in the deceased group but not LDH for the ICU group.

	Deceased vs. Emergency		ICU vs. Emergency	
	Area under the Curve (AUC)	95% CI for AUC	Area under the Curve (AUC)	95% CI for AUC
**AST**	0.592	0.484–0.7	0.485	0.362–0.607
**ALT**	0.784	0.693–0.876	0.804	0.703–0.906
**LDH**	**0.864**	0.781–0.946	0.428	0.303–0.553
**MGB**	**0.814**	0.734–0.895	**0.831**	0.744–0.919
**CK**	0.667	0.562–0.771	0.667	0.55–0.783
**CPR**	**0.975**	0.941–1	**0.972**	0.934–1
**Ferritin**	**0.882**	0.802–0.963	**0.808**	0.695–0.92
**D-dimer**	**0.914**	0.852–0.975	**0.872**	0.787–0.956

**Table 3 diagnostics-12-00176-t003:** Positive predictive values (PPV) in the deceased and ICU groups and negative predictive values (NPV) in the emergency group based on the upper reference value for AST, ALT, LDH, MGB, CK, CRP, Ferritin, and D-dimer (see methods). Ferritin data are expressed for men and women.

	AST 9–45 U/L	ALT 10–40 U/L	LDH 135–225 U/L	MGB 28–72 µg/L	CK 20–220 U/L	CRP 10–6000 µg/L	Ferritin 30–400 µg/L Men 15–150 µg/L Women	D-Dimer 50–420 µg/L
**PPV** **(Deceased *n* = 71)**	0.521	0.253	0.830	0.605	0.225	**0.957**	0.657 Men (*n* = 35) **0.944** Women (*n* = 36)	**0.915**
**PPV** **(ICU *n* = 54)**	0.222	0.166	0.240	0.722	0.074	**0.944**	0.461 Men (*n* = 26) **0.928** Women (*n* = 28)	**0.870**
**NPV** **(Emergency *n* = 31)**	**0.903**	**1.000**	0.709	**1.000**	**1.000**	**0.935**	0.750 Men (*n* = 12) **1.000** Women (*n* = 19)	0.096

## Data Availability

Data are available on request.

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
