# Peer review of "Early Routine Biomarkers of SARS-CoV-2 Morbidity and Mortality: Outcomes from an Emergency Section"

_diagnostics, 2022, doi:10.3390/diagnostics12010176_

Round 1
Reviewer 1 Report
Dear Editor,
We have thoroughly revised the manuscript entitled “Early routine biomarkers of SARS-CoV-2 morbidity and mortality. Outcomes from an emergency section” by Ceci FM, Fiore M, Gavaruzzi F et al. The study aims to evaluate the usefulness of routine blood biomarkers, like C-reactive protein, LDH, Ferritin, and D-dimer as potential early predictive factors of COVID-19 severity and death. The manuscript is well-structured, the ethics statements are appropriate and the conclusions are consistent with the results. Our main concern is the novelty of the results and conclusions, the role of these biomarkers as predictors of severe COVID-19 evolution having already been demonstrated.
Major issues
The main concern is the novelty of the results and conclusions, since several studies, systematic reviews, and meta-analyses have already described the role of each of these biomarkers as predictors of COVID-19 severity and outcome. The authors stated that “the strength of this investigation was to classify the COVID-19 patients according to their final prognosis instead of classifying patients using a score based on criteria correlated to severe COVID-19 disease” (lines 233-234). However, outcome-based evaluation is also used in other studies, including systematic reviews.
Minor issues
- A small number of minor corrections are needed
- Line 72 – “angiotensin-converting enzyme 2 (ACE2) receptor” instead of “angiotensin-converting enzyme 2 (ACE2)”
- Line 125 – RT-PCR instead of RT-CRP
- Line 126 – I would suggest “molecular test” instead of “molecular swab”
- Line 162 – “to analyze the laboratory data” instead of “to analyze the laboratory”
- It would be more appropriate to present the age range for each group (lines 119, 121, 123) in the Results section, instead of the Materials and Methods chapter.
- The phrase “The additives present in vacutainers are EDTA or sodium citrate as anticoagulants, and a gel to separate blood cells and blood serum” does not provide relevant information, since these are the standard blood collection tubes.
- The references for statistical analysis methods (lines 162-164) can be improved, references 21-23 are rather self-citations.
Kind regards,
Author Response
Reviewer 1
Dear Editor,
We have thoroughly revised the manuscript entitled “Early routine biomarkers of SARS-CoV-2 morbidity and mortality. Outcomes from an emergency section” by Ceci FM, Fiore M, Gavaruzzi F et al. The study aims to evaluate the usefulness of routine blood biomarkers, like C-reactive protein, LDH, Ferritin, and D-dimer as potential early predictive factors of COVID-19 severity and death. The manuscript is well-structured, the ethics statements are appropriate and the conclusions are consistent with the results. Our main concern is the novelty of the results and conclusions, the role of these biomarkers as predictors of severe COVID-19 evolution having already been demonstrated.
Reply. We thank the reviewer for the positive comments. We want also to stress the point that although many studies have been carried on the role of these biomarkers as predictors of severe COVID-19 evolution in hospitalized COVID-19 patients, to the best of our knowledge, no retrospective studies are available at the level of the emergency section only to early predict severe COVID-19 outcomes comparing 3 different groups of patients (those attending only the emergency section – those entering an emergency section and afterward attending the ICU for COVID-19 - those entering an emergency section and afterward attending the ICU but with a fatal outcome) (lines 259-277). We highlighted the changes we made in the text in light yellow.
Major issues
The main concern is the novelty of the results and conclusions, since several studies, systematic reviews, and meta-analyses have already described the role of each of these biomarkers as predictors of COVID-19 severity and outcome. The authors stated that “the strength of this investigation was to classify the COVID-19 patients according to their final prognosis instead of classifying patients using a score based on criteria correlated to severe COVID-19 disease” (lines 233-234). However, outcome-based evaluation is also used in other studies, including systematic reviews.
Reply. As suggested, we better explained the novelty of the present investigation (lines 259-277).
Minor issues
- A small number of minor corrections are needed
- Line 72 – “angiotensin-converting enzyme 2 (ACE2) receptor” instead of “angiotensin-converting enzyme 2 (ACE2)”
Reply. As requested, changes were made (line 72).
- Line 125 – RT-PCR instead of RT-CRP
Reply. As requested, changes were made (line 124).
- Line 126 – I would suggest “molecular test” instead of “molecular swab”
Reply. As requested, changes were made (line 125).
- Line 162 – “to analyze the laboratory data” instead of “to analyze the laboratory”
Reply. The statistical analysis section has been rewritten according to the comments of the 3 reviewers (lines 158-164).
- It would be more appropriate to present the age range for each group (lines 119, 121, 123) in the Results section, instead of the Materials and Methods chapter.
Reply. As suggested age range data were presented in the results (lines 167-169).
- The phrase “The additives present in vacutainers are EDTA or sodium citrate as anticoagulants, and a gel to separate blood cells and blood serum” does not provide relevant information, since these are the standard blood collection tubes.
Reply. As suggested the sentence has been deleted.
- The references for statistical analysis methods (lines 162-164) can be improved, references 21-23 are rather self-citations.
Reply. As requested, according to this comment we made changes citing references for ANOVA and ROC (lines 158-164).
Reviewer 2 Report
ANOVA was used, but did the Authors perform tests to assess normality? otherwise, non parametric analysis should be used. Statistical methods should be improdev, and their descriptin as well
Results: important data such as time elapsed from symptom onset to admission are missing. Also more data about deaths are required, and to be analyzed to the proposed markers
Furthermore, I think the authors identified interesting biomarkers that may correlate with clinical outcome of COVID-19. Hoever, further analysis on those biomarkers are envisaged (regression, ROC curves, combinations of the identified markers..)
Author Response
Reviewer 2
ANOVA was used, but did the Authors perform tests to assess normality? otherwise, non parametric analysis should be used. Statistical methods should be improdev, and their descriptin as well
Reply. As requested, the statistical analysis of the study was improved and rewritten. In particular, a receiver operating characteristic (ROC) analysis was performed to measure the diagnostic/predictive accuracy of each variable (data shown in Table 2 and Table 3). Normality was also assessed by Pearson's chi-squared test (Lines 158-164).
Results: important data such as time elapsed from symptom onset to admission are missing. Also, more data about deaths are required, and to be analyzed to the proposed markers.
Reply. We collected data at the level of an emergency section during the paroxysms initial phase of the pandemic and, unfortunately, the medical records of many patients were not filled in correctly, mainly for the emergency group (those patients attending only the emergency section) (lines 317-321).
Furthermore, I think the authors identified interesting biomarkers that may correlate with clinical outcome of COVID-19. Hoever, further analysis on those biomarkers are envisaged (regression, ROC curves, combinations of the identified markers..).
Reply. As suggested, ROC data were included. We thank the reviewer for the positive comments (lines 158-164; 228-238; 259-268).
Reviewer 3 Report
Authors attempted to identify routine blood biomarkers that might early indicate a severe COVID-19 progression in hospitalized patients. Information presented in this paper are important for the scientific community working in the field.
I have few comments:
Introduction:
Lines 101-103: “…the main aims and purposes of this retrospective study”? The aim of the study should be stated clearly, please rephrase. Also, to delete the word “dangerous”, I suggest the word “severe”.
Materials and Methods
Lines 119, 121, 123: please provide the mean age of patients
Lines 126-129: please rephrase, the exclusion criteria are not clearly presented.
Discussion
Lines 243-244: “Few studies have been conducted…”, authors should provide references for the statement
Line 276: authors to delete “of course”
Lines 275-279: please rephrase the study limitations, this is a long statement
Author Response
Reviewer 3
Authors attempted to identify routine blood biomarkers that might early indicate a severe COVID-19 progression in hospitalized patients. Information presented in this paper are important for the scientific community working in the field.
Reply. We thank the reviewer for the positive comments.
I have few comments:
Introduction:
Lines 101-103: “…the main aims and purposes of this retrospective study”? The aim of the study should be stated clearly, please rephrase. Also, to delete the word “dangerous”, I suggest the word “severe”.
Reply. As suggested, the sentence was rewritten (lines 101-103).
Materials and Methods
Lines 119, 121, 123: please provide the mean age of patients
Reply. As requested, we included the mean age of the patients (lines 167-169).
Lines 126-129: please rephrase, the exclusion criteria are not clearly presented.
Reply. According to the comments of the reviewer, the sentence was modified (lines 126-128).
Discussion
Lines 243-244: “Few studies have been conducted…”, authors should provide references for the statement
Reply. As suggested, we added additional references (lines 285-287)
Line 276: authors to delete “of course”
Reply. As requested “of course” was deleted.
Lines 275-279: please rephrase the study limitations, this is a long statement.
Reply. According to this suggestion, the limitations of the study were revised (317-321).
Round 2
Reviewer 2 Report
The manuscript has been imroved